# Skilful prediction of Sahel summer rainfall on inter-annual and multi-year timescales

K.L. Sheen[1,2], D.M. Smith[1], N.J. Dunstone[1], R. Eade[1], D.P. Rowell[1] & M. Vellinga[1]

Summer rainfall in the Sahel region of Africa exhibits one of the largest signals of climatic variability and with a population reliant on agricultural productivity, the Sahel is particularly vulnerable to major droughts such as occurred in the 1970s and 1980s. Rainfall levels have subsequently recovered, but future projections remain uncertain. Here we show that Sahel rainfall is skilfully predicted on inter-annual and multi-year (that is, > 5 years) timescales and use these predictions to better understand the driving mechanisms. Moisture budget analysis indicates that on multi-year timescales, a warmer north Atlantic and Mediterranean enhance Sahel rainfall through increased meridional convergence of low-level, externally sourced moisture. In contrast, year-to-year rainfall levels are largely determined by the recycling rate of local moisture, regulated by planetary circulation patterns associated with the El Niño-Southern Oscillation. Our findings aid improved understanding and forecasting of Sahel drought, paramount for successful adaptation strategies in a changing climate.

[1] Met Office Hadley Centre, Met Office, Fitzroy Road, Exeter, Devon EX1 3PB, UK. [2] Department of Physical Geography, University of Exeter, Peter Lanyon Building, Penryn Campus, Trelevier Road, Penryn, Cornwall TR10 9FE, UK. Correspondence and requests for materials should be addressed to K.L.S. (email: k.l.sheen@exeter.ac.uk) or to D.M.S. (email: doug.smith@metoffice.gov.uk).

The Sahel, which we define as the region between 16° W to 36° E and 10° N to 20° N, extends across the African continent, transitioning between two distinct climatological regimes: the Sahara desert to the north and the lush tropical rainforests to the south. Dry conditions pervade the Sahel throughout most of the year, with a wetter period between July and September associated with a northward shift of the main tropical rainfall band[1,2]. The Sahel summer rainy season exhibits one of the largest signals of global climatic variability[3], reflected in the fragility of agricultural productivity in north Africa. Prolonged drought ravished the region in the early 1980s, shattering crop and livestock farming and resulting in an estimated 100,000 deaths in rural communities from starvation, malnutrition and disease[4]. Sahelian summer rainfall has since recovered to the levels of the 1960s[3], but future projections are uncertain[5]. Superimposed on this multi-decadal change are substantial swings in inter-annual rainfall levels[2]. Understanding the physical drivers of both inter-annual and multi-year Sahelian rainfall variability is crucial to gain confidence in future predictions.

Climate models vary widely in their ability to capture Sahel rainfall anomalies. There is some statistical evidence of multi-year predictability, especially using multi-model means, for timescales greater than about 4 years and in the Western Sahel (Supplementary Table 1)[6–9]. No previous studies have documented predicting summer Sahelian rainfall from the previous winter, although seasonal predictions with lead times of 0–3 months have had considerable promise[10–17]. Model initialization is reported to improve both the amplitude and predictability of Sahel summer rainfall anomalies, although the results are model dependent[7,9,18,19]. Improved skill is also noted for models, which better simulate teleconnections between Sahel rainfall and sea-surface temperatures (SSTs)[20,21]. However, to our knowledge, no studies have clearly demonstrated skilfull predictions of the mechanisms by which global SSTs influence the moisture budget of the Sahel. Here we present improved understanding and prediction of Sahel rainfall variability on both multi-year and inter-annual timescales for lead times >8 months, using a comprehensive set of retrospective forecasts (hereafter hindcasts) from the latest Met Office Decadal Climate Prediction system, DePreSys3 (ref. 22). Covering the period since 1960, DePreSys3 uses a much higher resolution (~60 km in the atmosphere and 0.25° in the ocean) than previous versions[23,24], enabling better representation of the key physical processes[25] (Methods).

## Results
### Skilfull prediction of Sahel summer rainfall.
The DePreSys3 hindcasts skilfully predict post-1960 summer (that is, the mean of July, August and September) rainfall levels across the Sahel on both inter-annual and longer timescales (Fig. 1). For linearly de-trended mean summer rainfall averaged over the 2–5 year period after initialization (Year 2–5), correlations with observations are 0.73, 0.75 and 0.63 for the whole, west and east Sahel regions, respectively, and all are significant to the 95% level (see Methods). The west and east Sahel are delineated by 10° E. At shorter, 8 month leads, the skill of DePreSys3 for predicting the upcoming summers (Year 1) is more modest, yet still significant: $r = 0.48$, 0.50 and 0.33 for the whole, west and east Sahel regions, respectively. The drought period of the 1980s and the subsequent recovery are particularly well captured by the model, alongside some of the major inter-annual peaks and troughs in rainfall: for example in 1997–98 and 2012 (Fig. 1c,d). Accounting for ensemble size, DePreSys3 shows high skill compared with other initialized models and even the Coupled Model Intercomparison Project (CMIP5) multi-model mean hindcasts, in particular on

inter-annual timescales (Supplementary Table 1). However, understanding the physical mechanisms that modulate Sahel rainfall patterns is key for gaining further confidence in future predictions. We now assess these in detail using DePreSys3.

### Circulation patterns associated with Sahel rainfall change.
The Sahel sits at the confluence of several atmospheric circulation features that are illustrated schematically in Fig. 2. Examination of the summer climatological meridional flow reveals two areas of peak convergence and ascent (Fig. 3a): a deep circulation cell at 10°N, co-located with the Tropical Easterly Jet (TEJ) and the core of the tropical rain belt, and a shallower meridional overturning cell further to the north (~17°) that coincides with the surface manifestation of the Inter-tropical Convergence Zone (ITCZ)[2,26]. Zonally, Sahelian surface ascent peaks at the West African coast, just east of where the easterly Atlantic trade winds morph into the onshore West African Westerly Jet (WAWJ)[2,27] (Fig. 3b). This low-level flow, alongside a band of mid-atmospheric upwelling and upper level easterly flow spans much of the African continent. These atmospheric climatological patterns are consistent with several re-analysis products, although the deep circulation cell does not extend quite as far to the north in DePreSys3 (Supplementary Note 1 and Supplementary Fig. 3).

To distinguish the character of Sahelian rainfall drivers over different timescales, we analyse a large set of DePreSys3 1 year lead time hindcasts, separated into multi-year (that is, >5 years) and inter-annual timescales (Methods). Resultant timeseries are divided into wet and dry Sahel composites (Methods). Differences between wet and dry Sahel summer circulation patterns in DePreSys3 for both multi-year and inter-annual variability are shown in Fig. 3c–f. On multi-year timescales the deep convection cell is strengthened slightly and shifted northward by ~3.5° latitude during wet periods, displacing the main rainbelt further into the Sahel[28]. On inter-annual timescales, wet summers are modulated by a strengthened (as opposed to displaced) deep convection cell, coincident with a strongly enhanced zonal Walker-type circulation[29,30]. A poleward shift in the shallower circulation cell (that is, at 17°N) is apparent with increased rainfall on both timescales. As the lower cell meridional shifts however remain within the Sahel in DePreSys3, they will not impact domain-averaged rainfall levels. Similar analysis for re-analysis data are discussed in the Supplementary Note 1.

### Sahelian summer atmospheric moisture fluxes.
We next investigate changes in the moisture fluxed into the Sahel region at different atmospheric levels, that is, the summer mean specific humidity, $\langle q \rangle$ multiplied by the wind, $\langle \mathbf{u} \rangle$. Wet-minus-dry composites are shown in Fig. 4. Moisture flux changes are dominated by atmospheric levels below 400 hPa, reflecting the higher humidity content here. On both inter-annual and multi-year timescales, wetter summers are associated with increased moisture from the following: stronger northward winds between 600 and 850 hPa along the southern Sahel boundary; increased humidity carried by the Mediterranean sourced surface winds (even though they actually weaken slightly)—this is particularly important in the east of the region[31] Supplementary Fig. 7; and a stronger WAWJ. Moisture flux anomalies associated with the mid-altitude African Easterly Jet[32] shows a much stronger modulation on multi-year timescales and mostly counteract those from the lower level westerlies (Supplementary Note 2).

Analysis of column integrated moisture fluxes, $\int \langle q\mathbf{u} \rangle$, highlights the role of meridional circulations in modulating Sahel rainfall. The humidity of southward flowing surface winds from the Mediterranean and the strength of the northward monsoon flow dominate the overall moisture convergence in the Sahel,

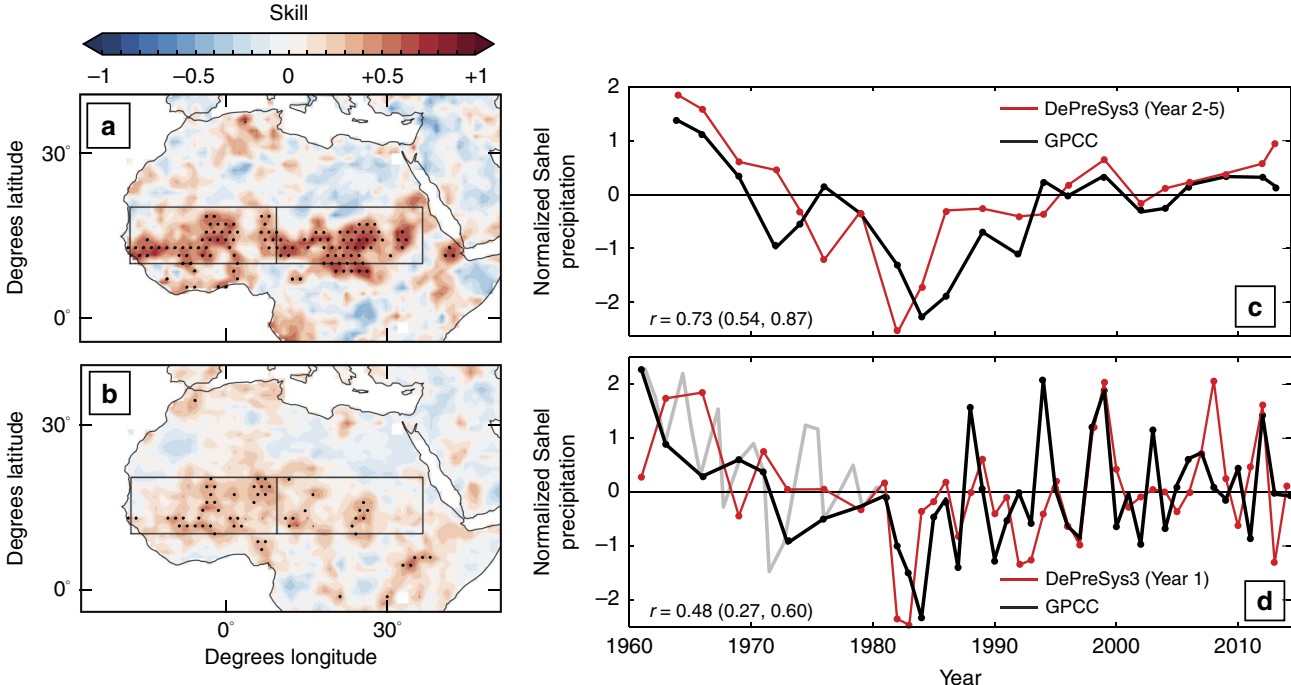

**Figure 1 | Skilful predictions of summer rainfall in the Sahel.** (**a**,**b**) Spatial distribution of anomaly correlation coefficient between DePreSys3 hindcasts of summer rainfall and equivalent GPCC observations on a 1° × 1° grid for 1960–2014. Boxes outline the Sahel region and delineate the west and east Sahel at 10° E. All time series are linearly detrended and normalized. Stippled regions mark 95% significance (see Methods). (**c**,**d**) Red line denotes the normalized and linearly detrended DePreSys3 ensemble mean timeseries of summer Sahel rainfall. Black lines mark observational rainfall series sub-sampled to match hindcast dates and the light grey line shows the full observational timeseries. Circles highlight each data point. Correlation coefficients with 5–95% confidence intervals are indicated in brackets (see Methods). Upper panels show results for summer rainfall data averaged over years 2–5 from initialization. Lower panels show individual seasons for hindcasts initialized in the previous November.

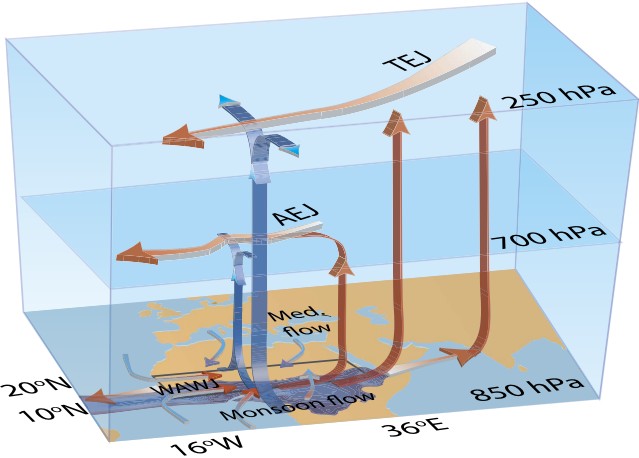

**Figure 2 | Schematic of climatological atmospheric features that modulate Sahel summer rainfall.** The blue region over Africa shows where observed July, August and September (JAS) rainfall levels are >2 mm per day on average. Arrows are schematic but based on summer climatology from DePreSys3 wind fields: red arrows show zonal Walker-type circulation at 250, 700 and 850 hPa; blue arrows mark meridional Hadley-type circulation and ascent. Key features associated with increased Sahel rainfall include: a stronger TEJ, which originates over the Tibetan Plateau and is also a response to local convective heating. The TEJ helps maintain the deep convection cell above the main rainband through cross-jet divergence[2,45]; the mid-level African easterly jet (AEJ), which sits north of the TEJ[32]; the WAWJ[27,34], which fluxes moisture from the Atlantic coast onto the north African continent; and meridional low-level convergence between the southerly monsoon winds and northerly winds of Mediterranean origin (marked Med. flow).

whereas little convergence is associated with the zonal WAWJ and African Easterly Jet components (Fig. 4). Changes in moisture convergence between wet and dry periods are similar on multi-annual and inter-annual timescales (0.17 and 0.16 mm per day, respectively, Fig. 4). However, the total rainfall change is higher on inter-annual timescales (0.28 mm per day compared with 0.23 mm per day for multi-year timescales). This difference is accounted for by a stronger contribution from evaporation (0.12 mm per day compared with 0.06 mm per day for multi-year timescales).

**External moisture supply versus local moisture recycling**. The relative roles of moisture advected into the region $P_a$ versus the recycling of local moisture $P_m$ are further assessed by computing timeseries of recycling ratios $\rho = P_m/P$, where $P = P_a + P_m$ is the total rainfall (see Methods)[33]. Sahel recycling ratios in DePreSys3 range from 0.3 to 0.4, suggesting that recycled water typically contributes just over a third of the total summer Sahel precipitation. However, we find a distinct contrast between the character of rainfall variability on different timescales: multi-year precipitation rate anomalies are weakly anti-correlated with the recycling ratio, whereas inter-annual fluctuations show a strong positive relationship: $r = 0.77(0.62, 0.88)$, where numbers in brackets represent the 5–95% confidence intervals. (Fig. 4). These relationships suggests the dominant role for dynamically driven local moisture recycling on inter-annual timescales, with external moisture supply being more important on multi-year timescales.

To further investigate the apparently different mechanisms driving Sahel rainfall change on different timescales, we compute the moist static energy (MSE), which enables the relative roles of moisture and temperature changes to atmospheric stability (and

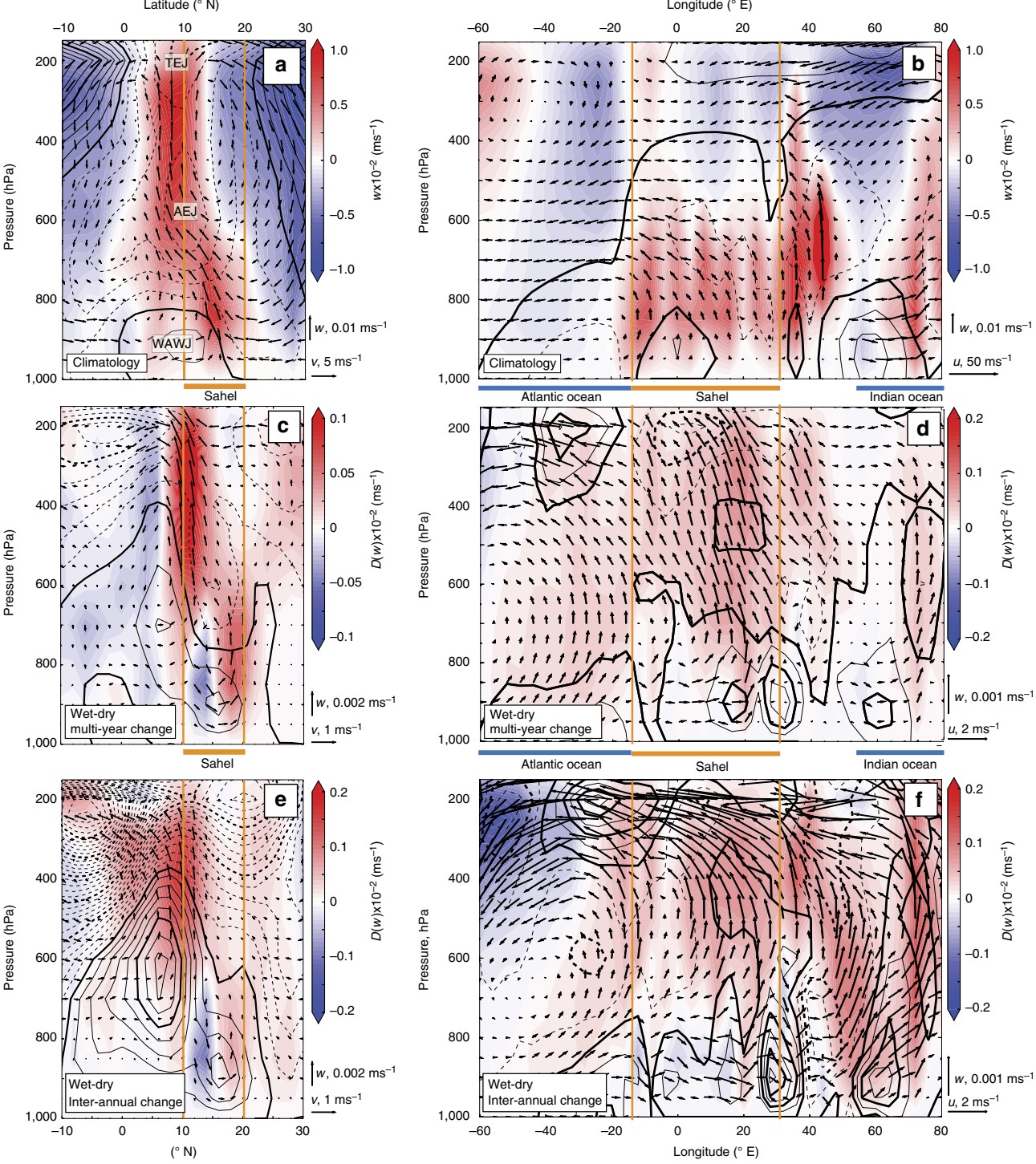

**Figure 3 | Atmospheric circulation associated with Sahel rainfall.** (**a**) Climatological situation. Black arrows show climatological meridional circulation across the Sahel. Colours represent vertical velocity and black thin (thick) contours show zonal velocities at 2 (10) ms$^{-1}$ intervals (dashed lines mark negative, easterly winds). All data are from the first year of DePreSys3 hindcasts and averaged between 16° W and 36° E. Wind vectors have been linearly interpolated onto a regular grid. The orange vertical bars delineate the Sahel domain. (**b**) As for **a** but for zonal circulation, such that wind vectors are averaged between 10° N and 20° N, and background thin (thick) contour lines show meridional velocity, spaced at 2 (4) ms$^{-1}$ intervals with dashed lines representing negative, northerly winds. (**c,d**) as for **a,b** but for wet minus dry composites for multi-year component of the one year lead time hindcasts. Thin (thick) contour spacings are 0.2 (1) ms$^{-1}$ for meridional winds and 0.125 (0.25) ms$^{-1}$ for zonal winds. (**e,f**) As for **c,d** but for inter-annual timescales.

hence ascent) to be assessed. The MSE represents the sensible, latent and geopotential energy, such that $MSE = c_p T + Lq + gz$, where $c_p$ is the specific heat of air at constant pressure, $T$ the temperature, $L$ the latent heat of vapourization of water, $q$ the specific humidity, $z$ the height and $g$ the acceleration due to

gravity[34]. An increasing MSE with altitude denotes a stable atmosphere. On multi-year timescales, increased moisture content ($Lq$) during wet years reduces the vertical stability throughout the atmosphere, but most intensely below 500 hPa (Fig. 5a). By comparison, on inter-annual timescales, it is the

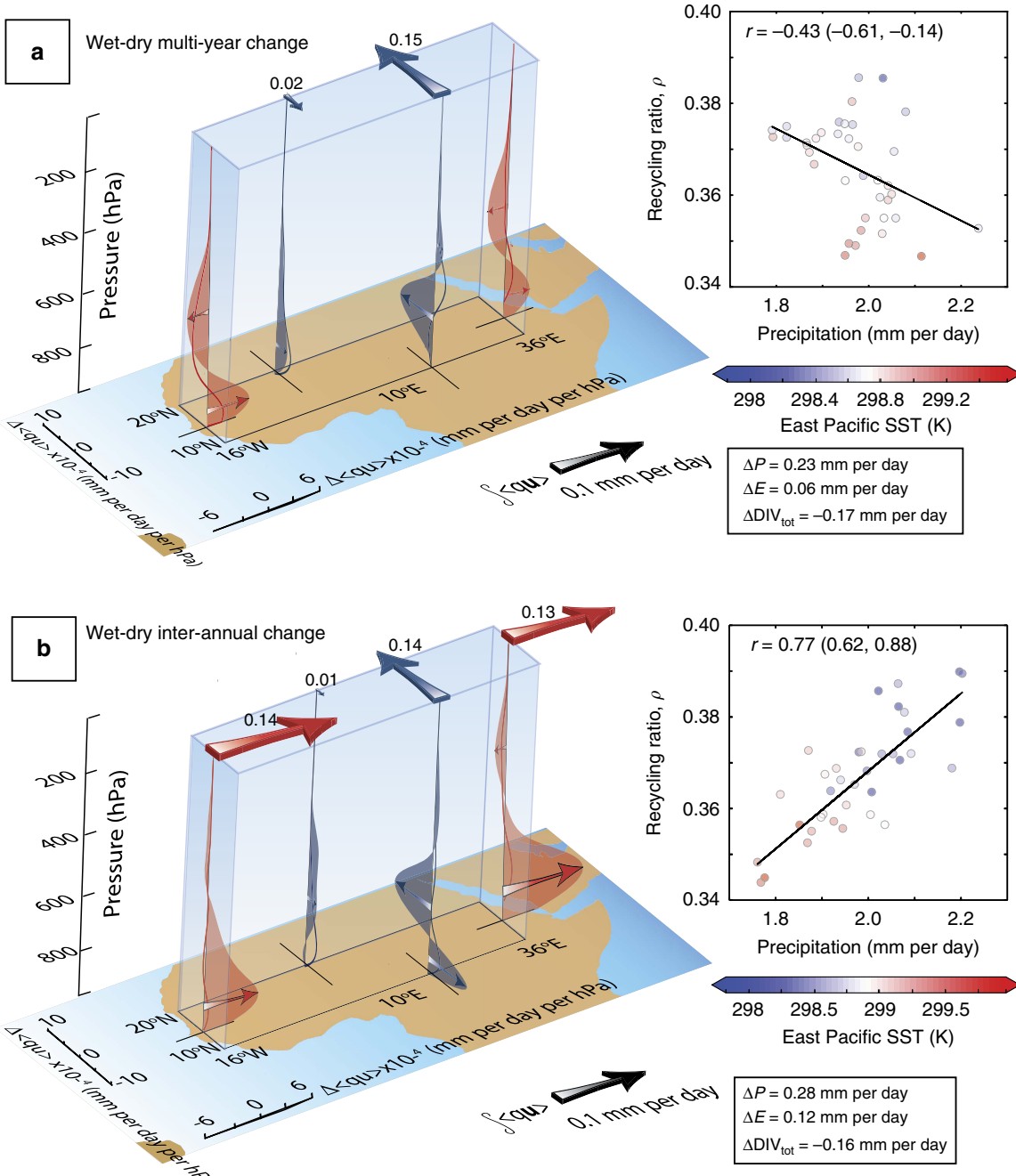

**Figure 4 | Changes in moisture between wet and dry Sahel summers in DePreSys3.** (**a**) Moisture flux changes for multi-year component of the one year lead time hindcasts. Red shaded regions show mean zonal moisture flux change between wet and dry years, $\Delta \langle q \rangle \langle u \rangle$, across the western and eastern limits of the Sahel as a function of pressure (see Methods). Red lines show the contribution from specific humidity changes, $\langle u \rangle \Delta \langle q \rangle$, as opposed to changes in the wind. Similarly, the blue regions and curves show meridional fluxes across the northern and southern edges of the domain. Thick red and blue arrows at top of the figure show changes in the zonal and meridional column integrated moisture fluxes, $\int \langle q \mathbf{u} \rangle$, with magnitudes indicated in mm per day (see Methods). The change in precipitation, $\Delta P$, evaporation, $\Delta E$, and moisture divergence, $\Delta DIV$, over the Sahel are also indicated. Right hand panel shows the relationship between precipitation and recycling ratio on multi-year timescales. Circle colours indicate an ENSO index: the mean temperature in the east Pacific region (120 to 170° W; − 5 to 5° N). The linear regression line and correlation values with 5–95% confidence limits are indicated. (**b**) As for **a** but for inter-annual component.

upper atmosphere (above 500 hPa), which is de-stabilized by cooler upper atmospheric temperatures ($c_{\mathrm{p}} T$) and therefore ascent in the deep convection cell enhanced, whereas at lower levels cooler temperatures counteract the increased moisture content (Fig. 5b). These results are consistent with the apparent changes in circulation patterns and moisture fluxes between wet and dry Sahel summers on different timescales (Figs 3 and 4):

meridional shifts in the circulation and moisture transport govern rainfall change on multi-year timescales, whereas upper-atmosphere temperature changes impact upper-atmosphere moisture ascent on inter-annual timescales.

**SSTs and teleconnections.** To gain further confidence in the skill of the forecasts, we now assess the global drivers of

moisture flux and local recycling that were found to be important on the two timescales. Several studies discuss the high sensitivity of the Sahel rainy season to the tropical and north Atlantic, the tropical Pacific, the Indian Ocean Dipole, the Mediterranean and the Saharan Low[3,18,20,31,35–42]. Similar results are found in DePreSys3: on multi-year timescales, a strong relationship exists between north Atlantic and Mediterranean SSTs and Sahelian summer rainfall (Fig. 6a), whereas the eastern tropical Pacific, eastern equatorial Atlantic and Indian Ocean are significantly correlated with Sahel precipitation inter-annually (Fig. 6b). Considering these teleconnections, we now return to consider the source of the skill in the year 2–5 and year 1 lead-time hindcast sets (Fig. 1). DePreSys3 is able to skilfully predict north Atlantic and Mediterranean SST on multi-year timescales, and

that of the tropical Pacific (that is, the El Niño-Southern Oscillation (ENSO)) and the western Indian Ocean (but not Indian Ocean Dipole) for year one hindcasts (Fig. 6c,d). However, we do not find significant skill in predicting the Saharan Low. Thus, we conclude that the north Atlantic and the Mediterranean on multi-year timescales, and the equatorial east Pacific and west Indian ocean on inter-annual timescales, are the fundamental sources of Sahel rainfall skill in DePreSys3 and evaluate the key mechanisms in more detail below.

On multi-year timescales, a comparatively warm north Atlantic acts to displace the marine ITCZ northward[24,43,44], shifting the tropical rainbelt and the associated meridional moisture convergence into the Sahel region (Figs 3c,4a and 7). Warmer northern hemisphere temperatures also lead to increased evaporation over the Mediterranean and the Atlantic Ocean, enhancing the specific humidity of air advected into the Sahel from these regions (Fig. 4a)[31]. In addition to elevating moisture convergence, this increased low-level moisture content over the Sahel acts to destabilize the atmosphere below 400 hPa, promoting convection and precipitation, particularly in the lower atmosphere (Fig. 5a). Similar patterns in multi-year circulation and MSE anomalies are seen for composites based on north Atlantic SST (25–60° N, 7–70° E; Supplementary Figs 8c,d and 9). In support of these processes underpinning the skilful prediction of multi-year rainfall change, we find that DePreSys3 year 2–5 hindcasts are able to forecast the precipitation, the low-level humidity and the wind anomalies associated with the sustained droughts of the 1970s and 1980s (Fig. 8). The equatorward migration of the marine ITCZ during the drought period is reflected in the north–south dipole pattern of rainfall anomaly maps (Fig. 8a,b)[38] and anomalous northerly winds fluxing moist air out of the Sahel across its southern boundary (Fig. 8c,d). Regions of reduced surface humidity associated with the drought are concentrated where winds originating from anomalously low humidity regions in the Mediterranean and Atlantic Ocean converge (Fig. 8e,f). Furthermore, we find that meridional shifts in the marine ITCZ, assessed as the latitude

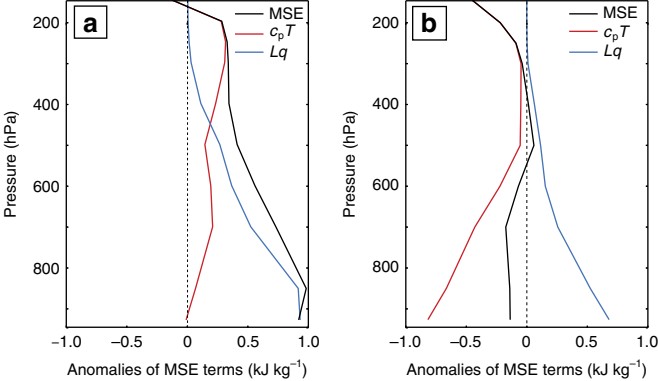

**Figure 5 | Drivers of atmospheric stability in DePreSys3.** Anomalies between wet and dry Sahel summers of MSE terms, averaged over the Sahel region. We ignore anomalies in the geopotential energy, which are found to be negligable. (**a**) Multi-year component of one year lead time hindcasts and (**b**) inter-annual component. An increasing MSE with altitude denotes a stable atmosphere.

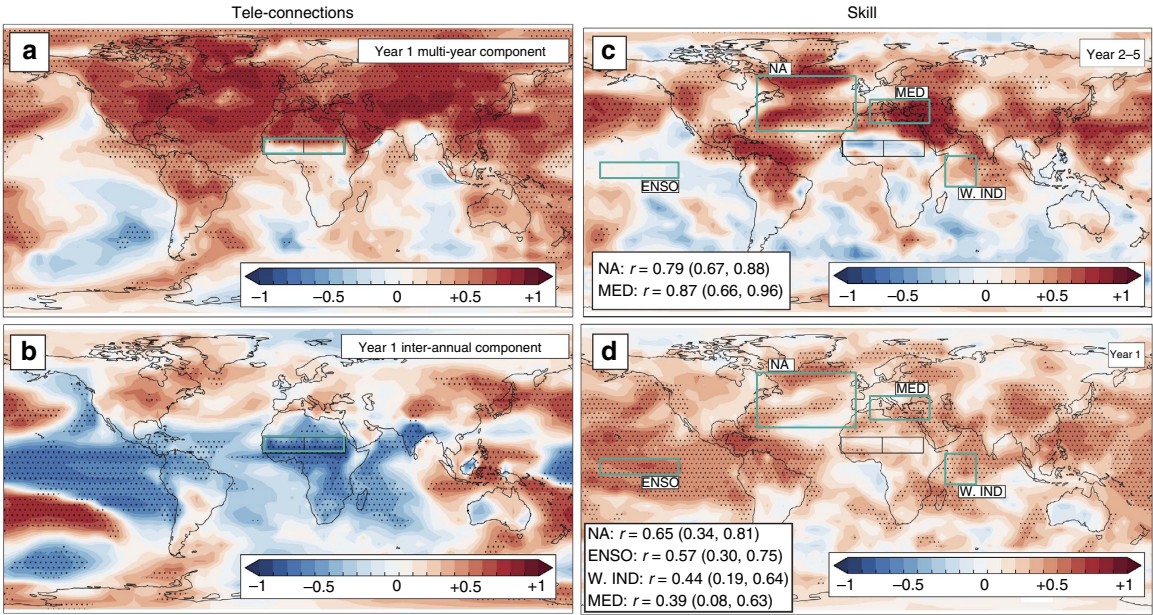

**Figure 6 | Relationship between global surface temperature patterns and Sahel rainfall.** (**a**) Correlation between Sahel summer rainfall and global surface temperatures for multi-year variability of year one lead time hindcasts (both detrended). (**b**) As for **a** but for inter-annual variability. (**c**) Skill of DePreSys3 July, August and September (JAS) detrended surface temperatures 2–5 years after initialization. Green boxes show ENSO index, North Atlantic (NA), west Indian Ocean (W. IND) and Mediterranean (MED) regions, the skill for which are given if significant. (**d**) As for **c** but for year 1 lead-time hindcasts. In all panels, correlations are computed in 2.5° × 2.5° boxes. Stippled regions mark correlations with significance at the 95% level (see Methods).

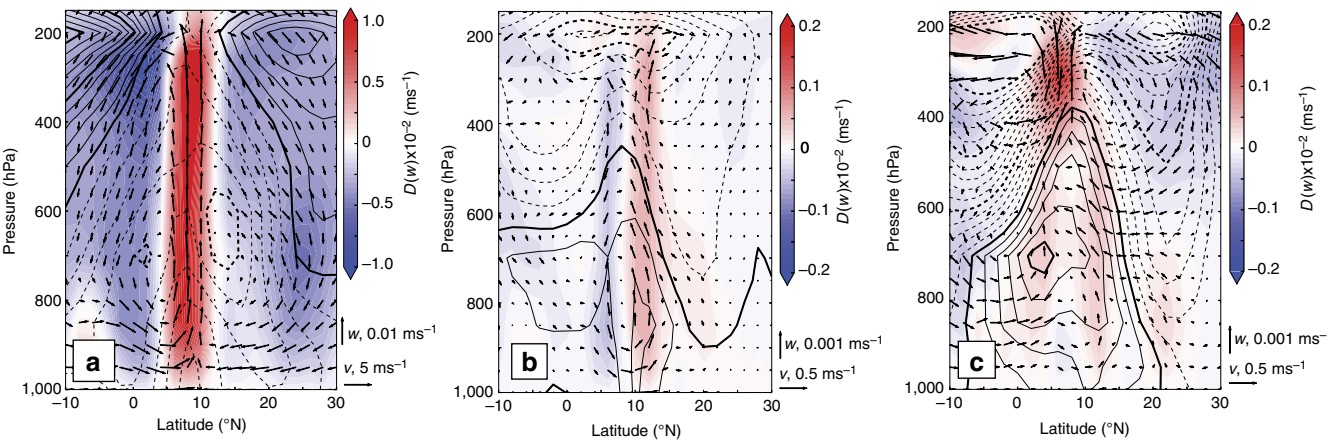

**Figure 7 | Anomalous circulation patterns of the marine ITCZ associated with Sahel rainfall change.** (**a**) As for Fig. 3a but for the tropical Atlantic Ocean: 16° W–50° W. Note the strong African Easterly Jet (AEJ) signal, marine ITCZ and lack of a shallow circulation cell. (**b,c**) As for Fig. 3c,e but for Atlantic ocean. The clear poleward shift in the deep convection cell on longer timescales and the strengthening of upper-level ascent on inter-annual timescales are noteworthy.

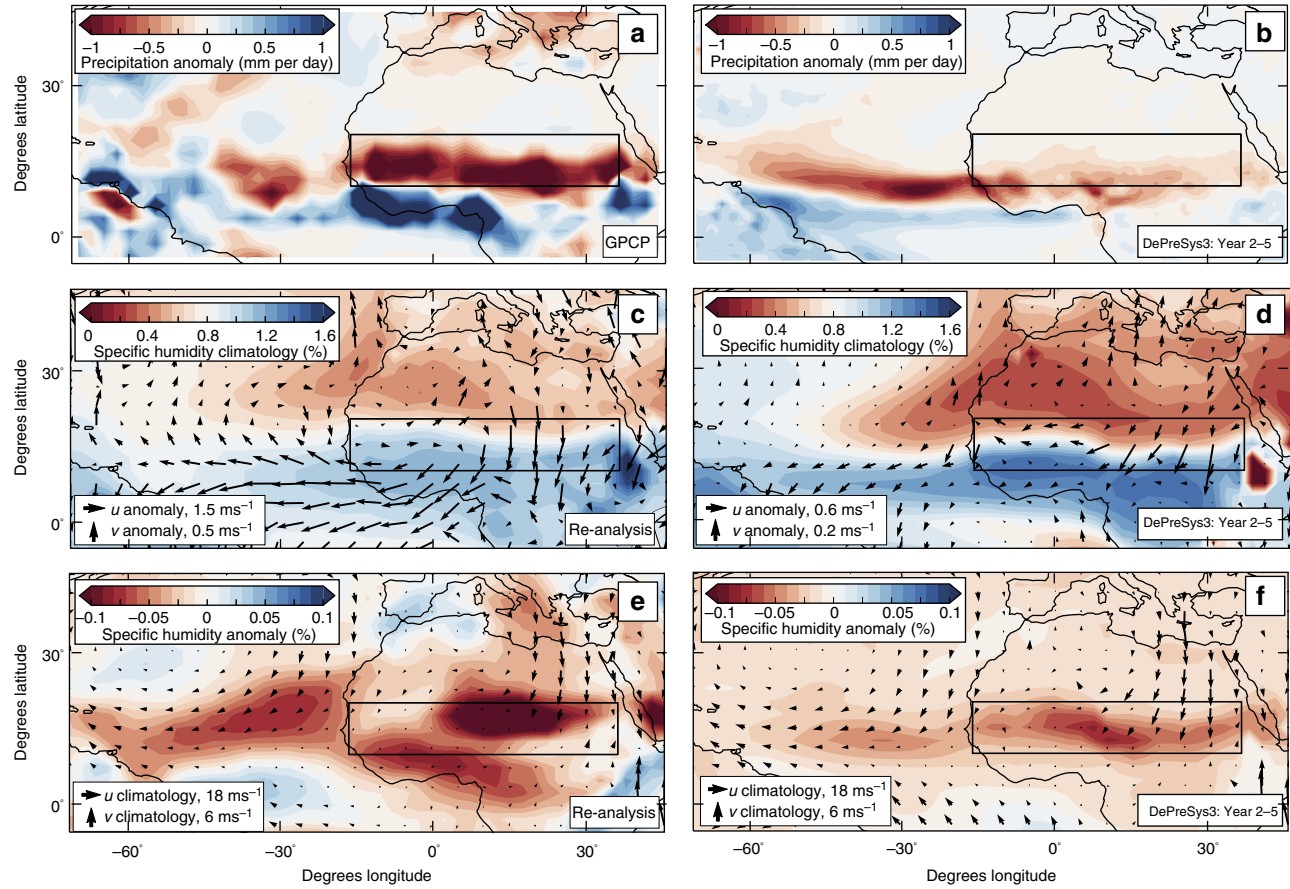

**Figure 8 | DePreSys3 predictions of the prolonged droughts of the 1970s and 1980s and the underlying mechanisms.** (**a**) Composites of observed precipitation anomaly over north Africa for the drought period (1975–1990) relative to other years post 1960. Included years are chosen to match DePreSys3 available members with lead times of 2–5 years—see Methods. Data have not been detrended or normalized such that it represent a true forecast. To show changes over the oceans we plot GPCP rainfall, although GPCC anomalies look similar over land. Black rectangle marks the Sahel. (**b**) Equivalent plot for DePreSys3 ensemble means, 2–5 years after intialization. (**c,d**) As for **a,b** but with colours representing the climatological specific humidity at 850 hPa and with arrows indicating the change in 850 hPa winds during the drought period (**c** is for re-analysis data). (**e,f**) As for **c,d** but with colours showing the specific humidity change at 850 hPa. Black arrows show 850 hPa climatological winds for period since 1960.

where 925 hPa meridional winds averaged between 50° W and 16° W reach zero magnitude, are skilfully predicted at the 95% level for year 2–5 hindcasts compared with re-analysis ($r = 0.57$).

Low-level (that is, 850 hPa) specific humidity changes over the Sahel are also skilful in the year 2–5 hindcasts, with a correlation of 0.89, significant at the 95% level.

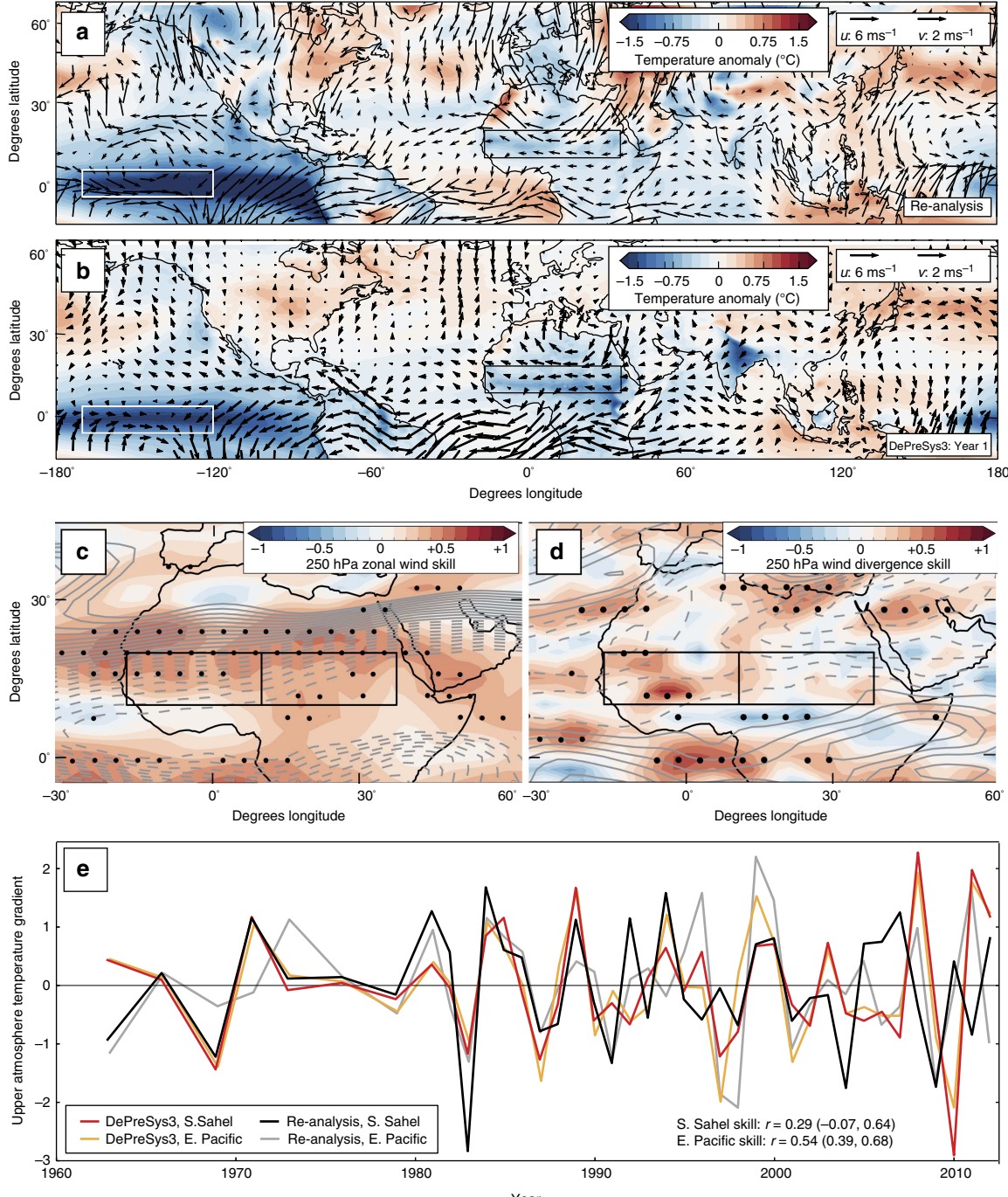

**Figure 9 | Skilful predictions of the mechanisms which modulate Sahel rainfall on inter-annual timescales.** (**a**) Inter-annual changes in 250 hPa winds (black arrows) and surface temperatures for re-analysis data based on cool minus warm south east Pacific SSTs (white box: the Nino 3.4 index region: −5 to 5° N, 120 to 170° W (ref. 69)). (**b**) As for **a** but for the inter-annual component of year one lead time DePreSys3 hindcasts. Similar patterns are seen for composites based on west Indian ocean SSTs. We also note that anomaly surface temperatures in the ENSO region are significantly correlated with the 250 hPa zonal wind strength ($r = 0.88$ for DePreSys3 and 0.65 for re-analysis) and with upper level divergence ($r = -0.57$ for DePreSys3 and −0.58 for re-analysis) in the Sahel region. (**c**) Skill map of 250 hPa zonal wind for inter-annual component. Grey contours indicate climatology and stippling represents 95% significance level for anomaly correlations. (**d**) As for **c** but for 250 hPa wind divergence. (**e**) Time series of upper-atmosphere temperature gradient ($T$ at 500 hPa minus $T$ at 200 hPa). The inter-annual component of DePreSys3 one year lead time hindcasts and equivalent data from re-analysis data are plotted. Indices are computed for the region associated with the deep convection cell at the Sahel, denoted by S.Sahel, (5 to 15° N, −16 to 36° E) and over the eastern tropical Pacific (white box in **a**). DePreSys3 skill levels for these time series are indicated. The strong connection between the upper atmosphere temperature gradient in the east Pacific and the south Sahel, suggesting a teleconnection mechanism: $r = 0.91(0.84, 0.96)$ and $0.40(0.24, 0.65)$ in DePreSys3 and the re-analysis, respectively, are also noteworthy. Furthermore, upper atmosphere temperature gradient in the Sahel is significantly correlated with Sahel precipitation: $r = 0.64$ for DePreSys3; (**b–e**) are for the first summer of DePreSys3 hindcasts initialized in the previous November.

The influence of ENSO and the western Indian Ocean on Sahelian rainfall is largely confined to inter-annual timescales (Fig. 6b). La Nina events are known to strengthen upper level easterlies (that is, the TEJ) across the Sahel through intensified surface temperature gradients, both zonally across the Pacific/Indian oceans and between the Tibetan Plateau and India[30,42,45] (Figs 3e,f and 9a,b). These upper level wind anomalies act to dynamically strengthen the deep convective upwelling cell in the Sahel through enhanced north–south wind divergence[2]. Furthermore, La Niña events and a cooler west Indian ocean destabilize the troposphere over the Sahel: anomalous downwelling and reduced latent heat release over the tropical Pacific and Indian oceans cool the global tropical upper atmosphere[46–48], resulting in reduced atmospheric stability between 200 and 500 hPa (Fig. 5b). Note also the cooler surface temperature across the Sahel associated with La Niña events (Fig. 9a,b), which acts to stabilize the lower atmosphere (Fig. 5b). Composites based on an ENSO index (SST in $-5$ to $5°$ N, $-170$ to $-120°$ W) or west Indian Ocean SSTs ($-10$ to $10°$ N, 50 to $70°$ W), produce similar patterns to Figs 3e,f and 5b (Supplementary Figs 8e,f and 9b). These two mechanisms, which are both skilfully predicted by DePreSys3 (Fig. 9c,d and e), amplify deep convection in the Sahel, leading to increased moisture recycling on inter-annual timescales (Fig. 4b).

## Discussion

We have shown that Sahel rainfall and the associated driving mechanisms are predictable on both multi-year and inter-annual timescales. However, we note that climatological rainfall levels and the amplitude of variability in DePreSys3 are too low, particularly poleward of $10°$ N (Fig. 10). Other modelling studies commonly report similar issues[12,49,50]. We also note that the

magnitude of moisture flux anomalies in the re-analysis data are generally greater than those in DePreSys3 (Fig. 5 and Supplementary Fig. 4).

We investigate the lack of rainfall in DePreSys3 by analysing the first empirical orthogonal function (EOF1) for Sahel precipitation, which accounts for $\sim 30\%$ of the variance in both the observations and DePreSys3. This analysis highlights that peak precipitation variability in DePreSys3 is about 2.5 (2) degrees too far south for hindcasts with year 2–5 (year 1) lead times (Fig. 10d,f). These EOF1 patterns also explain the sharp drop-off in correlation to the south of the Sahel (Fig. 1), which corresponds to opposite signs in the Global Precipitation Climatology Centre (GPCC) and DePreSys3 EOF1 patterns. To investigate whether the southward displacement of rainfall variability in DePreSys3 could account for the low mean and variance in modelled rainfall levels in the Sahel region, we re-compute rainfall timeseries for a box shifted by the appropriate degrees latitude, as evaluated using EOF1 patterns (Fig. 10d,f, orange boxes). We find that this shift accounts for the low climatological Sahel rainfall levels in DePreSys3, but does not wholly account for the low modelled variability. For example, for lead times of 2–5 years, the mean of each ensemble member standard deviation, $\langle \sigma_{ensem} \rangle$, is 0.16 for the shifted box compared to 0.31 for the Sahel GPCC timeseries. Furthermore, the ratio of predictable components[51] in observations relative to the model is greater than one (1.18) for DePreSys3, in agreement with results from CMIP5 multi-model ensemble[51]. This result suggests that Sahel rainfall is constrained too weakly by the predictable signal relative to the noise. Similar issues are seen for seasonal forecasts of the North Atlantic Oscillation[52]. Thus, the discrepancy in climatological rainfall levels is potentially explained by the main

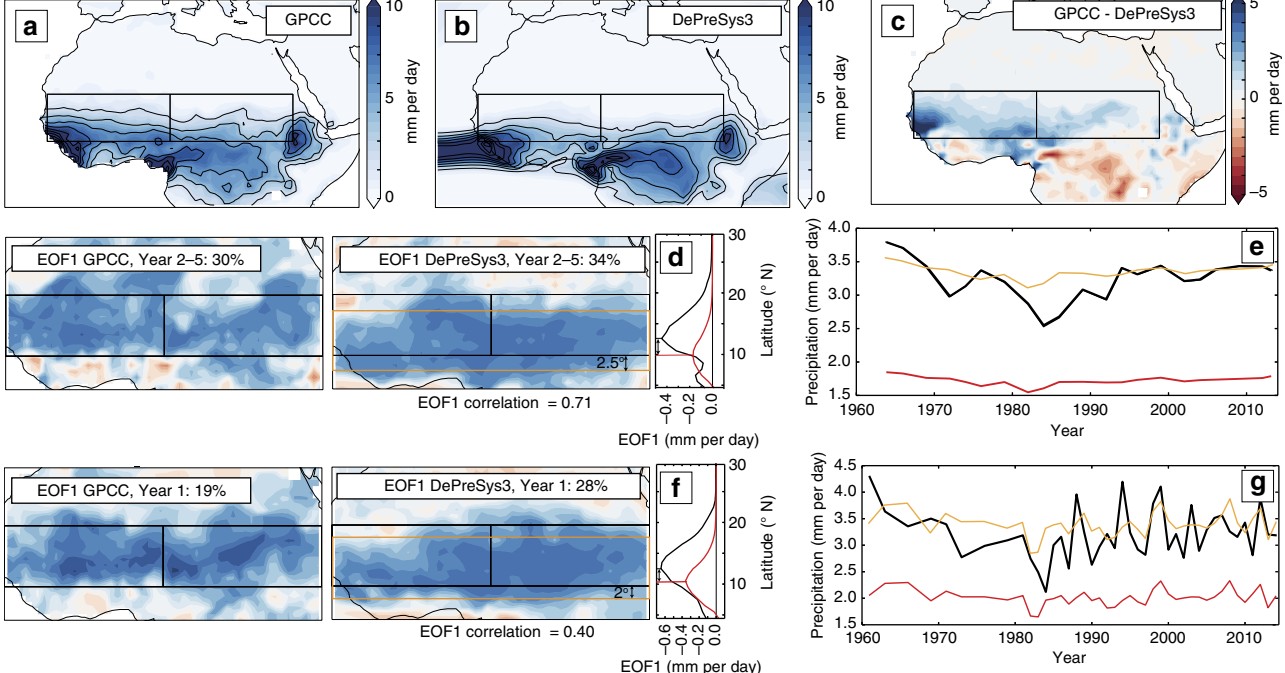

**Figure 10 | The climatology and variability of Sahel rainfall in DePreSys3 compared with observations. (a)** Mean observed summer rainfall from GPCC between 1961 and 2014. **(b)** As for **a** but for year one lead time DePreSys3 ensemble mean. **(c)** Difference between GPCC and DePreSys3 rainfall climatologies. **(d)** Maps show precipitation EOF1 pattern for DePreSys3 hindcasts with 2–5 year lead times and observations. The percentage of the variance accounted for by EOF1 and correlations between GPCC and DePreSys3 EOF1 principle components are indicated. Side panel shows EOF1 magnitude as a function of latitude across the Sahel for GPCC (black) and DePreSys data (red). Differences in latitudinal position of the EOF1 magnitude peaks are indicated with arrow. **(e)** Red line shows non-normalized but linearly detrended timeseries of DePreSys3 summer Sahelian rainfall for lead times of 2–5 years. Black line marks equivalent observational rainfall series. Orange line corresponds to DePreSys3 rainfall time series for region centered on EOF1 maximum (see orange rectangle in **d**). **(f,g)** As for **d,e** but for hindcasts with a one year lead time.

rainbelt in DePreSys3 not extending far enough north. However, even after correcting for this, the variability and signal to noise ratio of the model remains too low. We speculate that other factors, such as the Saharan Low, act to amplify Sahelian rainfall[36] in reality but are not represented properly in DePreSys3. Resolving these issues is beyond the scope of the this study, but could potentially lead to better skill in future.

In summary, this study demonstrates the ability of initialized climate models to skilfully predict and better understand both periods of sustained multi-year Sahel drought and year-to-year summer rainfall variability. Skilfull predictions of the underlying driving mechanisms increases our confidence that the high statistical skill represents physically realistic processes. On multi-year timescales, meridional moisture convergence plays a leading role in regulating Sahelian rainfall levels. Prolonged droughts are the result of equator-ward migration of the tropical rainbelt, weakened southerly monsoon winds and reduced low-level moisture content, in response to a cooler north Atlantic and Mediterranean Sea. On inter-annual timescales, the imprint of ENSO and western Indian ocean SSTs on tropospheric temperatures and zonal circulation modulates the ascent and recycling of local Sahel moisture through changes in upper-atmospheric stability and upper troposphere meridional wind divergence. These findings may also have implications for how the character of the Sahelian summer rains respond to global climate patterns: since tropical Pacific temperatures predominantly regulate the vertical stability and rate of local moisture recycling or threshold for deep convection, their impact may be largely manifest in the frequency of rainy days in the Sahel; north Atlantic and Mediterranean SSTs, on the other hand, may be expected to affect the median intensity of rainfall events through their modulation of moisture supply. In fact, data from Senegal indicates that the persistent droughts of the 1970s and 1980s, and the subsequent recovery, are associated with changes in the median intensity of daily rainfall events as opposed to number of rainy days[29,53], consistent with our expectations from multi-year changes driven by north Atlantic and Mediterranean SSTs. Our findings contribute to a deeper understanding of Sahel rainfall change and show that skilful predictions are possible months to years ahead. Such results are paramount to building resilience for adaptation to climate variability and change, improving future food security and economic stability in this vulnerable region.

## Methods

**DePreSys3.** DePreSys3 is based on the Hadley Centre Global Environment Model version 3, HadGEM3-GC2 (ref. 54) and is initialized by relaxing towards observed analyses of: ocean temperature and salinity from global covariance analysis[55] (nudged monthly with a 10-day relaxation timescales); sea-ice concentrations from the HadISST data set[56] (monthly with 24 h relaxation); and ERA-interim[57] atmospheric temperature and winds (nudged 6 h with 6 h relaxation). Data are assimilated from the surface and ramped up to 1 km altitude. The atmospheric resolution is 60 km with 85 quasi-horizontal atmospheric levels. The oceanic resolution is 0.25° with 75 levels. Hindcasts are fully influenced by external forcings (for example, greenhouse gases, aerosols, ozone, solar radiation and volcanoes) as per the CMIP5 protocol[58]. Ten ensemble members are run up to 16 months from initiation, for every year since 1980, and roughly every 2–3 years between 1960 and 1981 (1960, 1962, 1965, 1968, 1970, 1972, 1975, 1978 and 1980): 42 start dates in total. A smaller subset of these ten ensemble member model runs were continued to 5 years from intialization (1960, 1962, 1965, 1968, 1970, 1972, 1975, 1978, 1980, 1982, 1985, 1988, 1990, 1992, 1995, 1998, 2000, 2002, 2005, 2008 and 2009): 21 start dates. All members are initiated on 1 November. Time series are detrended by subtracting a linear trend line to focus on multi-year and inter-annual variability. We note that DePreSys3 precipitation estimates in the Sahel show deficiencies in capturing multi-decadal trends. A posteriori adjustment of trends is therefore required to make real-time predictions[59,60]. No other pre-processing was performed.

**Multi-year and inter-annual variability.** The multi-year component of year one lead-time hindcasts is extracted by smoothing time series with a 5-year running mean. The residual is used as the inter-annual signal. The time series are shown in Supplementary Fig. 2. Decomposing the Year 1 hindcasts in this way provides a fair comparison, because both multi-year and inter-annual time series have the same lead time and number of data points (this would not be the case if we compared the inter-annual component of the year 1 lead-time data with the hindcasts averaged 2–5 years from initialization). We do however note that moisture budget analysis performed on hindcasts averaged 2–5 years from initialization is largely consistent with that from the multi-year component of the Year 1 hindcast data (Supplementary Fig. 10). Resultant timeseries are divided into wet and dry Sahel composites by grouping wetter/dryer than average years and using a weighted average with weights determined by the absolute values of Sahel precipitation. In this way, all data points are included in wet and dry composites but very wet or dry years contribute more.

**Observations and re-analysis products.** Precipitation data were obtained from the monthly GPCC land-based dataset (1901–present)[61]. We note that results were similar when using the Climate Research Unit[62] precipitation data set and a subset of years with precipitation estimates from the Combined Precipitation Data Set (GPCP, available 1979–2015)[63]. The HadCRUT4 data set was used for global surface temperature estimates[64]. For winds, specific humidity and temperatures at all atmospheric levels, we use the NCEP/NCAR Reanalysis 1 project data provided by the NOAA National Center for Environmental Prediction[65]. Monthly means of daily means of variables were used. We do not include analysis, which requires the use of re-analysis precipitation and evaporation estimates, due to the large to uncertainties associated with these variables (Supplementary Fig. 3)[66]. For example, we found that the moisture budget in the Sahel did not close in the re-analysis data ($\Delta P - \Delta E \neq \Delta DIV$), stressing the limitations of diagnostic variables relating to the hydrological cycle in re-analysis products and the utility of model data in understanding Sahelian summer rainfall character.

**Moisture budget analysis and recycling ratio calculations.** Monthly means of specific humidity, $\langle q \rangle$, and wind, $\langle \mathbf{u} \rangle$, at different atmospheric levels are output by DePreSys3 (angled brackets represent the monthly mean). To evaluate the moisture flux contribution to the total Sahel region from each side of the domain as a function of atmospheric pressure, we average $\langle q \rangle \langle \mathbf{u} \rangle$ across the domain edge for each pressure level and divide by the domain length along the trajectory of the moisture flux[33]. In addition, moisture fluxes plotted as a function of pressure in Fig. 4, have been divided by $\rho_w g$, where $g$ is the acceleration due to gravity, and $\rho_w$ is the density of water. The result is to give units of mm per day per hPa. We note that estimates of $\langle q \rangle \langle \mathbf{u} \rangle$ represent the component of the total moisture flux, $\langle q\mathbf{u} \rangle$, due to the mean as opposed to the transient flow[67].

Exact column integrated moisture fluxes were computed during the model integration as $1/(\rho_w g) \int_0^{p_s} \langle q\mathbf{u} \rangle \mathrm{d}p$, where $\langle \rangle$ represents the monthly mean, $g$ is the acceleration due to gravity, $\rho_w$ is the density of water and $p_s$ is the surface pressure. Output of this integral is in units of m$^2$ s$^{-1}$, that is, it represents the flux of total moisture depth across each box edge. We average integrated fluxes across each domain edge and scale them by the length along the flux trajectory[33], such that units are in mm per day and are directly comparable to domain averaged values of precipitation and evaporation. We denote these scaled column integrated moisture fluxes as $\int \langle q\mathbf{u} \rangle$. The total moisture flux into, $F_{in}$, and out of, $F_{out}$, the Sahel are evaluated by summing the appropriate combinations of column integrated moisture fluxes, depending on their sign (that is, whether they represent flux advection into or out of the domain).

Recycling ratios, $\rho$, are evaluated as $P_m/P = E/(P + 2F)$, where $P_m$ is the contribution of precipitation due to the recycling of local moisture, $P$ the total precipitation, $E$ the evaporation and $F = 0.5(F_{in} + F_{out})$ is the average horizontal moisture flux per unit area due to the total flow. The recycling ratio is derived from moisture conservation using $P - E = F_{in} - F_{out}$ (ref. 33). Smaller values of $\rho$ indicate a dominance of the advective contribution to precipitate over the rate of recycling of local moisture.

**Assessing the significance of correlations.** To account for a finite ensemble size, $N_{ens}$, and finite number of points in each timeseries, $N_t$, on skill estimates of DePreSys3 ensemble mean outputs, we create an additional 1,000 hindcast time series as follows: Step 1: we randomly sample with replacement $N_{ens}$ ensemble members, from which we create an ensemble mean. Step 2: we resample and shuffle the resulting ensemble mean timeseries using block bootstrapping with a length of 4 years, being careful to account for irregular hindcast start times. The corresponding observational data points are reshuffled in the same way. Step 3: we compute the correlation between the re-shuffled ensemble mean and observational time series. Step 4: steps 1–3 are repeated 1,000 times. Correlation error bars for precipitation and surface temperatures as quoted in the text are computed as the 5–95% confidence interval limits from the resultant probability distribution of 1,000 correlation values. Stippling on maps represent grid points where >95% of the sampled hindcasts skill exceed zero[68]. For Sahel precipitation correlations shown in Fig. 1, we also evaluate whether the skill is significantly better than that which could be obtained by chance: we check that the skill exceeds the 95th percentile of a correlation distribution computed using the 1,000 hindcast sub-samples (as above) and the original unshuffled observation time series.

**Data availability.** The data that support the findings of this study are available from the corresponding authors upon request.

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

## Acknowledgements

This work was supported by the Joint DECC/Defra Met Office Hadley Centre Climate Programme (GA01101) and the EU FP7 SPECS project. The contribution of D.P.R. has received funding from the NERC/DFID Future Climate for Africa programme under the AMMA-2050 project, grant number NE/M019977/1. We are also grateful for some discussions with A. Scaife and C. Pomposi. We also thank two reviewers for their helpful suggestions.

## Author contributions

K.L.S. led the data analysis and writing, with suggestions and comments from all authors. K.L.S., D.M.S. and N.J.D. conceived the ideas for the work. N.J.D. set-up and ran the DePreSys3 hindcasts. R.E. analysed the Sahel precipitation skill of other CMIP5 models.

## Additional information

**Competing interests:** The authors declare no competing financial interests.

