## [Peer Review File · Nature Communications]

Reviewers' comments:

Reviewer #1 (Remarks to the Author):

Review of manuscript

Reference: NCOMMS-16-03828

Title: Skilful prediction of Sahel summer rainfall on inter-annual and multi-year timescales

Authors: K. L. Sheen; D. M. Smith, N.J. Dunstome, R. Eade, D.P. Rowell; M. Vellinga

General:

The authors analyse the skill in predicting Sahel rainfall with the DePreSys3 system in two time scales: interannual and multi-year. They show statistical significant skill in both and research into the possible causes of such skill. Their analyses suggest that wet minus dry years over the Sahel are related to increased convergence of moisture, mainly dominated by the meridional circulation on both time scales. There is, however, a difference in terms of local moisture recycling, which seems to play the dominant role on inter-annual time scales but not on multi-decadal ones. The authors relate this to the destabilization of the upper part of the tropospheric column on inter-annual time scales due to cold anomalies there, while on multi-annual time scales the whole column is destabilized due to increased moisture content imported into the region. They also show that the global drivers of such changes are SST anomalies: north Atlantic and Mediterranean warm anomalies on multi-year timescales and cold eastern tropical Pacific and Indian ones on interannual promote more rainfall over the Sahel and are skilfully predicted by the system.

In general I find the paper very long (it has 8 figures in the main text) and complex to follow. There are many detailed analyses, which is nice, but maybe it would be more suited for a long paper rather than a letter.

Novelty and interest of the paper:

In the introduction the authors point out 2 main novelties of their research:

A. Long-range prediction of interannual Sahel rainfall (i.e. 8 months) has not been reported before.

B. No previous study has clearly demonstrated skilful predictions of the mechanisms by which global SSTs influence the moisture budget of the Sahel.

As for A, I think it could be of great interest to a wide community, not only climate scientists. However, I also think that the authors give little revision on the previous works related to seasonal predictions of Sahel rainfall. Most of the citations of works that evaluate predictions are focus on multi-year predictions (references 7, 9, 10 and 11; references 6 and 8 do not evaluate predictions).

Regarding B, I find it is a very detailed analysis and I'm not sure, aside from the climate research people, how many audience will be interested in the details of the moisture budget of the Sahel in the DePreSys3 predictions. Conversely, I think it will make a very nice paper in a climate orientated journal.

In addition, I think the comparison with observations is a must if one wants to claim prediction of the (observed) mechanisms: In the manuscript, Fig. 3 is not confronted with obs, only Fig. S3 but I understand that this is again model outputs, right? I can't find a Figure 5 for observations, neither one for Fig. 6ab. For those figures that show an observation counterpart, there are some differences that are not discussed. For instance, Fig 4 and S2, the magnitudes of the anomalies are much bigger in the observations (see axes for qu anomalies). The meridional moisture flux across the northern boundary of the Sahel seems quite different between obs and model. Why? Can't recycling be estimated in obs/reanalysis?

In Fig. 7cd the anomalous winds in the model underestimate those in the observations, and in Fig. 7 of the specific humidity anomalies are not that similar over the Sahel. In Fig. 8 the increase in the TEJ in reanalysis is not clear.

Further comments:

-I find it confusing the multi-year data description: are there two ways to look at multi-year data? On the one hand the 2-5 year averages in Fig. 1ac and on the other the yr1 multi-year smoothing using 5 year means as explained in the Methods section? When I read the captions of the figures that lead with "wet-dry multi-year change", I'm not sure if I'm looking at one type or the other. And if it is the latter, I'm not really sure what I'm looking at. How is this 1yr smoothed data related to the prediction skill in the 2-5 years?

-In Table 1, Fig. 1, Fig. S1, why do you de-trend the data? How do you de-trend the data? Can this detrending be used in real-time prediction (i.e. is it a measure of real skill)? What are the scores if you don't de-trend it?

-Regarding the composites, it would be nice to know where the anomalies are statistically significant.

-Regarding the statistical significance of correlations, I've read the section several times, but I wouldn't be able to replicate it if I were given your data. Could you explain it a bit more? I'm confused about step ii). With what do you correlate the re-sampled and shuffled time series to build the correlation error bars? Regarding the significance of Sahel precipitation tested against the null hypothesis of correlations coming by chance, do you apply it to multi-year data that is filtered with a 5-yr running mean? Do you think it convenient?

-What about testing if DePreSys3 beats persistence in the 2-5 year prediction?

-I find fig. 4 difficult to evaluate quantitatively. I acknowledge that the plot is quite intuitive but I wouldn't be able to provide a quantitative measure of fluxes if I was asked to. The angles make it complicated in the 3D plot.

-Could you explain a bit better the sentence in line 175-177: it is not clear at what height the north-south divergence would be (high levels?). Also the temperature gradients (also at high levels?), where do they point to? How are they related to the divergence?

-In the methods section, when describing the DePreSys simulations that have been continued to 5 years, are there also 10 members for each of the 21 start dates ?

-What is the reason of the selection of the white area in the Pacific to build the warm-cold composites?

-It would be very nice if you could provide longitudes and latitudes axes and even grids in the plots with maps (like Fig. 1ab, Fig. 6-8 and S4).

-Authors are missing in the reference of Rowell et al. (2016).

-I think the correct year for reference 13 is 2013.

-I think caption of Fig. S3 is wrong, the figure you are referring to is Fig. 4, right? Not Fig. 4a.

Reviewer #2 (Remarks to the Author):

Skilful prediction of Sahel summer rainfall on inter-annual and multi-year timescales

A. Summary of the key results

This study uses a updated version of a decadal climate prediction system to demonstrate the skillful prediction of Sahel rainfall on interannual and multi-year (>5 year) timescales. In addition, using the decadal climate prediction system, the authors were able to illustrate the different mechanisms driving precipitation variability at the two timescales and that these mechanisms were also skillfully predicted in the model system.

I returning the manuscript to the authors for minor revisions.

B. Originality and interest: if not novel, please give references

The examination of mechanisms of Sahel rainfall using a decadal prediction system is novel and will be of interest to decadal prediction system developers and seasonal forecasters, among others.

C. Data & methodology: validity of approach, quality of data, quality of presentation

The paper is well presented. The majority of the analysis uses GPCP data - how much does the skill vary when using other long-term precipitation datasets such as CRU? The methodological approach is valid. I was unsure if any pre-processing (e.g. removing model drift) of DePreSys occurred as is typical in analysis of decadal hindcasts. Please clarify.

D. Appropriate use of statistics and treatment of uncertainties

Statistics and uncertainty are well presented in the manuscript and utilized effectively.

E. Conclusions: robustness, validity, reliability

The conclusions are clear, concise, and valid. The results could be reproduced with other decadal prediction systems given the information in the manuscript.

F. Suggested improvements: experiments, data for possible revision

- The climatological circulation and wet-dry composites are presented for the DePreSys system (Fig. 2). How similar are these to the reanalysis products used and results from prior studies that have looked at interannual and multi-year circulations (they are referenced earlier but linking to these results would be beneficial). This analysis would help solidify the validity of the system used.
- How much does skill vary if a different reanalysis is used for the mechanistic analysis?
- Be clear in the results (line 58) that you are referring to the earlier discussion of long-term droughts I incorrectly assumed to be the decadal hindcasts in CMIP5 (as opposed to the historical or future runs). With the earlier discussion of long-term droughts I incorrectly assumed you were considering the decadal scale (e.g. >10 years) or you were considering the decadal scale (e.g. >10 years).
- Clarify earlier in the manuscript that multi-year refers to >5 years. With the earlier discussion of long-term droughts, I incorrectly assumed you were considering the decadal scale (e.g. >10 years).
- In lines 106-108 you mention that the rainfall changes are larger on interannual scales because evaporation contributes more. Why? This seems to contradict the statement in lines 155 that states that the large-scale warming in the multi-year periods leads to more evaporation.
- The labels (a,b,c,d) in Fig. 4 are not consistent with the caption
- Figure 8d is not discussed in the text.
- Supplementary figure 1c shows multi-year "decadal" in S1a, which is inconsistent with the remainder of the paper.

G. References: appropriate credit to previous work?

Appropriate credit is given in the introduction but additional comparison in the results section (particularly in the circulation patterns section) would be beneficial.

H. Clarity and context: lucidity of abstract/summary, appropriateness of abstract, introduction and conclusions

Well written, logical, and concise.

RESPONSE TO REVIEWER COMMENTS

Reviewer #1 (Remarks to the Author):

General:

The authors analyse the skill in predicting Sahel rainfall with the DePreSys3 system in two time scales: interannual and multi-year. They show statistical significant skill in both and research into the possible causes of such skill. Their analyses suggest that wet minus dry years over the Sahel are related to increased convergence of moisture, mainly dominated by the meridional circulation on both time scales. There is, however, a difference in terms of local moisture recycling, which seems to play the dominant role on inter-annual time scales but not on multi-decadal ones. The authors relate this to the destabilization of the upper part of the tropospheric column on inter-annual time scales due to cold anomalies there, while on multi-annual time scales the whole column is destabilized due to increased moisture content imported into the region. They also show that the global drivers of such changes are SST anomalies: north Atlantic and Mediterranean warm anomalies on multi-year timescales and cold eastern tropical Pacific and Indian ones on interannual promote more rainfall over the Sahel and are skilfully predicted by the system.

1. In general I find the paper very long (it has 8 figures in the main text) and complex to follow. There are many detailed analyses, which is nice, but maybe it would be more suited for a long paper rather than a letter.

Thanks for the suggestions. We have modified several aspects of the manuscript in order to clarify the text and hope that it now follows more of an article style paper, rather than a letter. As suggested by the editor we have expanded our references and included some of the Supplementary Information (Section 5 and Section 6 from the original manuscript) into the main article. We believe that this has improved the flow of the manuscript and made it more accessible for the reader to follow. We do however note that Reviewer 2 describes the manuscript as ‘Well written, logical and concise’.

2. Novelty and interest of the paper:

In the introduction the authors point out 2 main novelties of their research:

A. Long-range prediction of interannual Sahel rainfall (i.e. 8 months) has not been reported before.

B. No previous study has clearly demonstrated skilful predictions of the mechanisms by which global SSTs influence the moisture budget of the Sahel.

As for A, I think it could be of great interest to a wide community, not only climate scientists. However, I also think that the authors give little revision on the previous works related to seasonal predictions of Sahel rainfall. Most of the citations of works that evaluate predictions are focus on multi-year predictions (references 7, 9, 10 and 11; references 6 and 8 do not evaluate

predictions).

We have now included several more references related to seasonal predictions of Sahel rainfall (lines 35-36). We have removed references 6 and 8 from this section. We have also include some more key references in line 143 : Zhang and Delworth, 2006 and Knight et al., 2006.

3. Regarding B, I find it is a very detailed analysis and I'm not sure, aside from the climate research people, how many audience will be interested in the details of the moisture budget of the Sahel in the DePreSys3 predictions. Conversely, I think it will make a very nice paper in a climate orientated journal.

We appreciate the Reviewers' comment, but we do believe that *Nature Communications* is an appropriate journal to publish this work. The scope of *Nature Communications* is 'to represent important advances of significance to specialists within each field, such that we wanted to include more specialist, but highly relevant, analysis (e.g. the moisture budget), whilst still appealing to a more general audience. Using clear diagrammatic explanations, we have ensured that the manuscript is both accessible and of interest the non-specialist. As mentioned by the reviewer, we believe that the work will be of great interest to the general community, such that *Nature Communications* is an appropriate channel to disseminate our findings. Furthermore, the more generous content allowance of *Nature Communications*, allows us to present our detailed analysis in an 'article' as opposed to 'letter' format as advised.

4a. In addition, I think the comparison with observations is a must if one wants to claim prediction of the (observed) mechanisms: In the manuscript, Fig. 3 is not confronted with obs, only Fig. S3 but I understand that this is again model outputs, right?

To address this concern, we have analysed two re-analysis products to produce a version of Fig. 3, as requested by the reviewer. The products that we have used are the Japanese 55 year analysis (JRA-55) and the NCEP/NCAR reanalysis products. In our previous manuscript we used both the ERA-40 (available 1957-2002) and ERA-INTERIM data (covers years post-1979). This analysis involved joining two timeseries together which introduced spurious discontinuities. To avoid this we now use the NCEP/NCAR and JRA-55 products (both available 1960-2016).

Firstly, we find that the atmospheric climatology is largely consistent between DePreSys3 and the re-analysis products (i.e. Figure 3a & 3b in main text, and Figures Aa & Ab below). One subtle difference is the position of the African Easterly Jet (AEJ), which sits slightly further to the north in the re-analysis. In addition, the deep circulation cell tends to extend further to the north in the re-analysis, supporting our discussions around Figure 10 in the manuscript. Due to this climatological northward shift, the re-analysis data show enhanced ascent at upper levels in the zonal circulation section when compared to DePreSys3 (Fig. 3b and Ab). We have now included this discussion along with Figure A in the Supplementary Information, Section 3 and note in lines 76-79: 'These atmospheric climatological patterns are consistent with several re-analysis products, although the deep circulation cell does not extend quite as far to the north in DePreSys3 (Supplementary Information, Section 3).'

Examination of the re-analysis wet minus dry composites show that Sahel rainfall is associated with similar circulation shifts as in DePreSys3: on multi-year timescales the meridional circulation pattern is migrated northward during wet years and the AEJ and WAWJ strengthen; on inter-annual timescales, the deep convection cell is strengthened and the zonal Walker cell and TEJ enhanced. However, we do note that differences in the wet minus dry circulation patterns between the different timescales are not as distinct as in the DePreSys3 data. In particular, the re-analysis data indicates that in addition to a strengthening, there is also a hint of a meridional migration of the Sahelian deep circulation cell on inter-annual timescales.

To produce the wet and dry composites for these plots, we decided to use the GPCP precipitation timeseries, rather than the precipitation output from the re-analysis. Our rationale for this came from a closer examination of the precipitation output from different re-analysis products (Figure B). It is clear that there are large inconsistencies between different re-analysis products and the GPCP observations, and as noted on the NCAR website: ‘Diagnostic variables relating to the hydrological cycle, such as precipitation and evaporation, should be used with extreme caution’ (<https://climatedataguide.ucar.edu/climate-data/atmospheric-reanalysis-overview-comparison-tables>). Furthermore, we find that the moisture budget does not close in re-analysis data, as also shown by Seager and Henderson (2013), as referenced in the Methods section.

The uncertainty in precipitation and evaporation estimates from re-analysis products highlights the utility of models such as DePreSys3, and it is also why we have chosen to only base DePreSys3 skill metrics on wind and humidity re-analysis variables in the main manuscript. We have also decided that it is best not to include results that incorporate re-analysis precipitation or evaporation. We have commented on this in the Methods section with appropriate references (lines 288-293), and have also included Figure B in the Supplementary Information (Section 3) to justify this decision.

Figure A: As for Figure 3 in main text, but for the average of NCEP/NCAR and JRA-55 products and with wet and dry composites determined using the GPCC observational precipitation data.

Figure B: Timeseries of Sahel summer precipitation from GPCCC observations and three different reanalysis products. Data has not been de-trended

4b. I can't find a Figure 5 for observations, neither one for Fig. 6ab.

Versions of these figures are now included in the Supplementary Information (Fig. S5 & S6), along with a short discussion (Section 3).

We have also updated the moisture budget re-analysis figure for the NCEP/NCAR and JRA-55 datasets (now Fig. S4).

5. For those figures that show an observation counterpart, there are some differences that are not discussed. For instance, Fig 4 and S2, the magnitudes of the anomalies are much bigger in the observations (see axes for q_u anomalies). The meridional moisture flux across the northern boundary of the Sahel seems quite different between obs and model. Why?

As pointed out by the reviewer, although the patterns look broadly similar, moisture flux differences are more pronounced in the observations. In part, we refer to our comment above regarding the limitations of the re-analysis products. In addition, as previously noted in the Supplementary Material (Section 4), column integrated moisture fluxes from the re-analysis data are imperfect, as we were limited to computing the column integral of $\langle q \rangle \langle u \rangle$ (where $\langle \rangle$ represents the monthly mean) as

opposed to the column integral of $\langle q \rangle$. $\langle q \rangle$ represents the contribution to moisture flux changes of the monthly mean circulation and unlike the integral of $\langle q \rangle$, do not encompass variability due to the intra-monthly eddy flow [Pomposi, 14]. However we do agree that in general, DePreSys3 does not exhibit enough rainfall variability or exhibit as intense wet-minus-dry moisture flux anomalies as re-analysis data. We address this issue by including the following sentences in the main text (lines 200-202): ‘We also note that the magnitude of moisture flux anomalies in the re-analysis data are generally greater than those in DePreSys3 (Figs. 5 & S4).’ We also discuss this issue in Section 3 of the Supplementary Material, lines 71 - 75.

Clearly there is room for improved skill in the model, particularly if the magnitude of the variability, and southward shift of the main rainband in DePreSys are addressed. These issues are addressed in the Summary and Discussion section where we highlight the differences between observations, re-analysis and model data.

6. Can't recycling be estimated in obs/reanalysis?

We have decided not to include estimates involving precipitation and evaporation observations in the paper as these metrics are not well constrained (see points above). We now note this in the caption to figure S3.

7. In Fig. 7cd the anomalous winds in the model underestimate those in the observations, and in Fig.7 of the specific humidity anomalies are not that similar over the Sahel.

We have resolved these issues by re-plotting this figure using the NCEP-NCAR dataset, which has a continuous record from 1960 to 2014 and so does not involve merging two different re-analysis products together as was done in the previous submitted manuscript. We also find a better correlation between DePreSys3 multi-year lower level specific humidity variability and the NCEP-NCAR product. Please see amended Figure 8.

8. In Fig. 8 the increase in the TEJ in reanalysis is not clear.

Again, utilising the NCEP-NCAR product, as opposed to joining the ERA-40 and ERA-interim datasets has largely resolved this point. Please see new Figure 9.

Further comments:

9. I find it confusing the multi-year data description: are there two ways to look at multi-year data? On the one hand the 2-5 year averages in Fig. 1ac and on the other the yr1 multi-year smoothing using 5 year means as explained in the Methods section? When I read the captions of the figures that lead with "wet-dry multi-year change", I'm not sure if I'm looking at one type or the other. And if it is the latter, I'm not really sure what I'm looking at. How is this 1yr smoothed data related to the prediction skill in the 2-5 years?

Yes, to analyse the multi-year signal we use both 2-5 year averages and the yr1 smoothed data. The motivation behind this approach is as follows. In reality, forecasts can only be produced for a full timeseries that contain contributions from the different frequency components. Throughout the paper we therefore assess the skill of DePreSys3 using the full (non-decomposed) timeseries as plotted in Figure 1 (i.e. averaged 2-5 years from initialization). By comparison, when investigating the mechanisms that drive rainfall change on different timescales (e.g. moisture fluxes, moist static energy, sea surface temperatures), we wanted to distinctly separate the inter-annual and multi-year components and thus chose to decompose the yr1 timeseries into different frequency components. This method also enabled a fair comparison between multi-year and interannual variability, seeing that both timeseries

had the same lead-time and number of data points, which would not be the case if we compared the inter-annual component of the yr1 data with the 2-5 year averages. We therefore use the decomposed yr1 timeseries for this part of the analysis. We have checked that the mechanisms driving the smoothed component of the yr1 timeseries are similar to those in the yr 2-5 lead time averaged data. As an example we include the moisture flux anomaly diagram for the yr 2-5 timeseries below (Figure E). We have clarified this in the Methods Section (lines 271-275).

Figure E: As for Figure 5a in main manuscript but for DePreSys3 hindcasts, averaged 2-5 years from initialization (Yr 2-5)

10. In Table 1, Fig. 1, Fig. S1, why do you de-trend the data? How do you de-trend the data? Can this detrending be used in real-time prediction (i.e. is it a measure of real skill)? What are the scores if you don't de-trend it?

As noted in line 52 and Figure caption 1, the data is linearly de-trended. We de-trend the data to focus on predicting multi-year and inter-annual changes, which are of interest to many potential users. The long term trend simulated by DePreSys is actually greater than that seen in the observations, and this discrepancy in trends reduces the correlations to 0.32 and 0.42 for multi-year and inter-annual timescales, respectively. However, differences between modelled and observed trends are not uncommon in decadal predictions, and approaches for adjusting the trend by post processing have previously been developed [e.g. Kharin et al 2012, Fuckar et al., 2014, as now referenced in Methods section] so that real-time predictions can be made.

To clarify we have included the following sentence in the methods section (lines 263-267): ‘Time series are de-trended by subtracting a linear trend line in order to focus on multi-year and inter-annual variability. We note that DePreSys3 precipitation estimates in the Sahel show deficiencies in capturing multi-decadal trends. A posteriori adjustment of trends is therefore required in order to make real-time predictions [Kharin et al. 2012; Fuckar et al., 2014].’

11. Regarding the composites, it would be nice to know where the anomalies are statistically significant.

Such statistical significance tests are paramount when assessing predictive skill. However, we do not think that they are necessary for the more qualitative, descriptive assessments of changes associated with Sahel rainfall change, and would clutter the figures. We do however assess the skill of the driving mechanisms to add rigour to our analysis.

12. Regarding the statistical significance of correlations, I've read the section several times, but I wouldn't be able to replicate it if I were given your data. Could you explain it a bit more? I'm confused about step ii). With what do you correlate the re-sampled and shuffled time series to build the correlation error bars? Regarding the significance of Sahel precipitation tested against the null hypothesis of correlations coming by chance, do you apply it to multi-year data that is filtered with a 5-yr running mean? Do you think it convenient?

We apologise that this section is not clear and have clarified step (ii) in the Methods section (lines 321-327). We do not apply the null-hypothesis correlations to the year one lead time data that is smoothed with a running mean: this approach is only used for the un-decomposed timeseries as plotted in Figure 1. We have clarified this in the Methods section (lines 331-332).

13. What about testing if DePreSys3 beats persistence in the 2-5 year prediction?

Thanks for the suggestion. We have computed the correlations for DePreSys3 documented in Table 1 in the Supplementary Information but for persistence i.e. using each DePreSys3 prediction to forecast for the subsequent observation. Persistence skill results are as follows for the year 2-5 and year 1 lead times:

JAS Y2-5	Whole Sahel	West Sahel	East Sahel
Persistence	0.50	0.49	0.40
JAS Y1	Whole Sahel	West Sahel	East Sahel
Persistence	0.21	0.34	0.03

We have now included these results in Table 1 in the Supplementary Information.

14. I find fig. 4 difficult to evaluate quantitatively. I acknowledge that the plot is quite intuitive but I wouldn't be able to provide a quantitative measure of fluxes if I was asked to. The angles make it complicated in the 3D plot.

This figure is supposed to be largely qualitative and give a broad overview of the key atmospheric features which impact rainfall in the Sahel on difference timescales. We feel that adding more scaling will complicate the figure and make it less accessible to a general readership. We have however included the actual integrated flux numbers in Figure 4 so that it can be interpreted more qualitatively.

15. Could you explain a bit better the sentence in line 175-177: it is not clear at what height the north-south divergence would be (high levels?). Also the temperature gradients (also at high levels?), where do they point to? How are they related to the divergence?

We have changed these lines to read:

‘La Nina events strengthen upper level easterlies (i.e. the TEJ) across the Sahel through intensified

surface temperature gradients both zonally across the Pacific / Indian oceans and between the Tibetan Plateau and India. These upper level wind anomalies act to dynamically strengthen the deep convective upwelling cell in the Sahel through enhanced upper level north-south wind divergence (Figs. 3e, 3f, 8a & 8b).’

16. In the methods section, when describing the DePreSys simulations that have been continued to 5 years, are there also 10 members for each of the 21 start dates ?

Yes there are 10 members for each of the start dates. We have clarified this in the Methods section in line 261.

17. What is the reason of the selection of the white area in the Pacific to build the warm-cold composites?

This is the region that we have used throughout the paper as our ENSO metric, and is based on the box used for the NINO 3.4 index. We have now included the appropriate reference in the figure caption. We use this as our index to build composites to illustrate the impact on ENSO variability on upper level winds (see lines 181-184). ‘La Nina events are known to strengthen upper level easterlies (i.e. the TEJ) across the Sahel through intensified surface temperature gradients both zonally across the Pacific / Indian oceans and between the Tibetan Plateau and India’

18. It would be very nice if you could provide longitudes and latitudes axes and even grids in the plots with maps (like Fig. 1ab, Fig. 6-8 and S4).

We have added the axes to Figs. 1a, 1b, 8 and 9, although we don’t think it is required for the full world maps and Fig. S6 is already quite busy (the Sahel region is marked with black rectangles however for location reference).

19. Authors are missing in the reference of Rowell et al. (2016).

I think the correct year for reference 13 is 2013.

I think caption of Fig. S3 is wrong, the figure you are referring to is Fig. 4, right? Not Fig. 4a.

Thanks for pointing these errors out – they have now been corrected

Reviewer #2 (Remarks to the Author):

Review of Manuscript: NCOMMS-16-15110

Skilful prediction of Sahel summer rainfall on inter-annual and multi-year timescales

A. Summary of the key results

This study uses a updated version of a decadal climate prediction system to demonstrate the skillful prediction of Sahel rainfall on interannual and multi-year (>5 year) timescales. In addition, using the decadal climate prediction system, the authors were able to illustrate the different mechanisms driving precipitation variability at the two timescales and that these mechanisms were also skillfully predicted in the model system.

I am returning the manuscript to the authors for minor revisions.

B. Originality and interest: if not novel, please give references

The examination of mechanisms of Sahel rainfall using a decadal prediction system is novel and will be of interest to decadal prediction system developers and seasonal forecasters, among others.

Thankyou

C. Data & methodology: validity of approach, quality of data, quality of presentation

The paper is well presented. The majority of the analysis uses GPCP data - how much does the skill vary when using other long-term precipitation datasets such as CRU? The methodological approach is valid. I was unsure if any pre-processing (e.g. removing model drift) of DePreSys occurred as is typical in analysis of decadal hindcasts. Please clarify.

Thanks for the suggestions. We have analysed the skill for the CRU dataset as suggested. Equivalent results as shown in Table 1 for GPCP data set are as follows:

JAS Y2-5	Whole Sahel	West Sahel	East Sahel
DePreSys3	0.75c	0.72c	0.67c
JAS Y1	Whole Sahel	West Sahel	East Sahel
DePreSys3	0.43c	0.53c	0.22c

c = significantly better than climatology at 90% level (or indeed at the 95% level)

If anything skill generally seems improved slightly for the CRU data set. We had previously checked correlations with the GPCP dataset, as included in the Methods of the original manuscript, but we now also mention the CRU data set in the Methods: line 283.

No pre-processing of the data was performed. We did not deem it necessary to remove model drift because all ensembles used have the same lead time and correlations will not be sensitive to model drift. We have now clarified this in the Methods section (line 267).

D. Appropriate use of statistics and treatment of uncertainties

Statistics and uncertainty are well presented in the manuscript and utilized effectively.

Thank you.

E. Conclusions: robustness, validity, reliability

The conclusions are clear, concise, and valid. The results could be reproduced with other decadal prediction systems given the information in the manuscript.

Thank you

F. Suggested improvements: experiments, data for possible revision

- The climatological circulation and wet-dry composites are presented for the DePreSys system (Fig. 2). How similar are these to the reanalysis products used and results from prior studies that have looked at interannual and multi-year circulations (they are referenced earlier but linking to these results would be beneficial). This analysis would help solidify the validity of the system used.

Please see our response to Reviewer 1 above (point 4).

We also now include further references to other studies which have looked at these circulations in this section: Nicholson, 2008; Grist & Nicholson, 2001 and Pomposi et al, 2016. However, we could not find any studies that directly compare multi-year and inter-annual circulation wet minus dry anomalies in this way.

- **How much does skill vary if a different reanalysis is used for the mechanistic analysis?**

The skills computed in our previous manuscript included both the ERA-40 And ERA-INTERIM data. At the suggestion of the reviewer we have re-computed the skill for the key mechanism using the NCEP/NCAR re-analysis, which we now use in the revised manuscript, as this does not involve joining two different re-analysis products together. Results for the NCEP/NCAR data are consistent:

On multi-year timescales, meridional shifts in the marine ITCZ, are still skilfully predicted at the 95% level for year 2--5 hindcasts when using the NCEP/NCAR re-analysis ($r = 0.57$). Furthermore we find that low-level (850 hPa) specific humidity changes over the Sahel are also skilful in the multi-year hindcasts, with a correlation of 0.89, significant at the 95% level.

On inter-annual timescales, results for the NCEP/NCAR re-analysis are now displayed in the new Figure 9.

- **Be clear in the results (line 58) that you are referring With the earlier discussion of long-term droughts I incorrectly assumeto the decadal hindcasts in CMIP5 (as opposed to the historical or future runs).**

We have clarified this to read: ‘Accounting for ensemble size, DePreSys3 shows high skill compared to other initialized models and even the Coupled Model Intercomparison Project (CMIP5) multi-model mean **hindcasts**, particularly on inter-annual timescales (see Supplementary Information, Section 1)’

- **Clarify earlier in the manuscript that multi-year refers to >5 years. With the earlier discussion of long-term droughts, I incorrectly assumed you were considering the decadal scale (e.g. >10 years).**

We have now clarified ‘multi-year’ in the abstract (line 10).

- **In lines 106 -108 you mention that the rainfall changes are larger on interannual scales because evaporation contributes more. Why? This seems to contradict the statement in lines 155 that states that the large-scale warming in the multi-year periods leads to more evaporation.**

We apologise for the confusion. Inter-annual changes in precipitation are larger in the analysis, and this increased change compared to multi-year variability is largely accounted for by increased evaporation as opposed to external moisture advection, as stated in lines 109-113. The increased evaporation on multi-year timescale in original line 155, referred to evaporation over the Atlantic and Mediterranean, the source of the air which eventually gets fluxed over the Sahel. We have now clarified this in lines 160-162 and lines 172-174.

- **The labels (a,b,c,d) in Fig. 4 are not consistent with the caption**

Corrected

- **Figure 8d is not discussed in the text.**

We now reference Fig 8d (which is now 9d) (line 193)

- Supplementary figure 1 calls multi-year "decadal" in S1a, which is inconsistent with the remainder of the paper.

We have changed to 'decadal' to 'multi-year' in the Figure S2

G. References: appropriate credit to previous work?

Appropriate credit is given in the introduction but additional comparison in the results section (particularly in the circulation patterns section) would be beneficial.

Please see our response to point F above.

H. Clarity and context: lucidity of abstract/summary, appropriateness of abstract, introduction and conclusions

Well written, logical, and concise.

Thank you.

REVIEWERS' COMMENTS:

Reviewer #1 (Remarks to the Author):

Reference: NCOMMS-16-03828

Title: Skilful prediction of Sahel summer rainfall on inter-annual and multi-year timescales

Authors: K. L. Sheen; D. M. Smith, N.J. Dunstome, R. Eade, D.P. Rowell; M. Vellinga

This is the second review of the manuscript. After the author's changes I think the paper reads much better, I was able to follow it more easily. I also think it is a valuable contribution worth publishing. Most of the concerns raised in the last revision have been addressed by the authors and I just have two comments, which could be considered a minor revision, and a few very minor details.

My first concern was already raised in the last review: the fact that you analyse the yr 1 smoothed (5yr running mean) to research into the mechanisms underlying the good skill obtained in predicting Sahel rainfall at lead time yr 2-5. I disagree with the authors' response in that the data averaged 2-5 years from initialization contain contributions from the different frequency components. In the process of averaging results from years 2-5 you are low-pass filtering with a running mean (length 4 years). So in Fig. 1c you are looking at multi-year phenomena. I understand that there are several reasons for your choice of analysing yr 1 smoothed data, as you explained in the response. However, I think that it is relevant for your paper to at least mention somewhere in the text that the results with the yr 1 smoothed data (Figs. 3cd, 4a, 5a, 6a) are consistent with the ones obtained with yr 2-5 data. This is so because one of the highlights of your paper is showing "skilful predictions of the mechanisms by which global SSTs influence the moisture budget of the Sahel".

My second concern is related to the comparison with reanalysis. I wonder if it makes more sense to compare the model's results with reanalysis but using the same years / loads in the composites as in the model, this is, not using the GPCP precipitation time series to produce the wet and dry composites but the model's one. In this way you would be checking that the predicted mechanisms agree with the "observed" ones for those events that can be predicted by the system. You would remove from the possible causes of the mismatches the fact of using an independent dataset to determine the composites.

Details:

-line 10 "... multi-year (i.e. \sim 5 years) timescales ...". I think it would be better to put ">" rather than " \sim ". The yr 2-5 average you use is like a running mean with a window length of 4 years, which is a low-pass frequency filter with an approximate cut-off period of 9 yrs.

-line 62-64: "The fact that several models...robust over the historical period". I'm not sure what this means: You were not sure about the statistical assessment you were using but once you applied and some models showed skill you are more confident in it? I would remove this sentence.

-line 222: "... such as the Sahel Low..". Do you mean Saharan Low?

-Fig 3c: Make the colorbar the same as in the rest of wet minus dry composites for easier comparison.

-Fig 5: where these calculated in the Sahel box? Please, clarify.

-Fig. 9: line 630: where is the upper level divergence calculated? Is ENSO also significantly anti-correlated with TEJ strength and upper level divergence in reanalysis? The sill map in plot c) is anomaly correlation? Please clarify.

Supplementary Information:

-Caption in table 1, last sentence: What do you mean by "Persistence skill is also included for DePreSys3? Typically persistence is calculated using observations, right? At least that was what I

tried to suggest in the last review.

-Line 127: "In DePreSys3, we find that the AEJ does not ..." This would seem also the case for the reanalysis (Fig. S4), right?

Reviewer #2 (Remarks to the Author):

The revisions made by the authors have substantially improved the paper. The clearer discussions, additional links to prior results, and testing other reanalysis and precipitation datasets improves the validity of the conclusions.

RESPONSE TO REVIEWER COMMENTS

We thank both reviewers for taking the time to read our manuscript again. Whilst Reviewer 2 recommended publication without any further work, Reviewer 1 provided some further helpful minor revisions that have been dealt with in the following way.

Reviewer Response

Reviewer #1 (Remarks to the Author):

Reference: NCOMMS-16-03828

Title: Skilful prediction of Sahel summer rainfall on inter-annual and multi-year timescales

Authors: K. L. Sheen; D. M. Smith, N.J. Dunstone, R. Eade, D.P. Rowell; M. Vellinga

This is the second review of the manuscript. After the author's changes I think the paper reads much better, I was able to follow it more easily. I also think it is a valuable contribution worth publishing. Most of the concerns raised in the last revision have been addressed by the authors and I just have two comments, which could be considered a minor revision, and a few very minor details.

Thank you

My first concern was already raised in the last review: the fact that you analyse the yr 1 smoothed (5yr running mean) to research into the mechanisms underlying the good skill obtained in predicting Sahel rainfall at lead time yr 2-5. I disagree with the authors' response in that the data averaged 2-5 years from initialization contain contributions from the different frequency components. In the process of averaging results from years 2-5 you are low-pass filtering with a running mean (length 4 years). So in Fig. 1c you are looking at multi-year phenomena. I understand that there are several reasons for your choice of analysing yr 1 smoothed data, as you explained in the response. However, I think that it is relevant for your paper to at least mention somewhere in the text that the results with the yr 1 smoothed data (Figs. 3cd, 4a, 5a, 6a) are consistent with the ones obtained with yr 2-5 data. This is so because one of the highlights of your paper is showing "skilful predictions of the mechanisms by which global SSTs influence the moisture budget of the Sahel".

The reviewer raises an important point that we have addressed by including the

following sentence in the revised manuscript (lines 273-275) 'We do however note that moisture budget analysis performed on hindcasts averaged 2-5 years from initialization, is largely consistent with that from the multi-year component of the Yr1 hindcast data (not shown).'

We also agree with the reviewer's point that the data averaged 2-5 years will not contain contributions from inter-annual variability. Considering this we have removed this comment from the article such that original line 272 in the Methods section has been changed from 'Decomposing the Yr1 hindcasts in this way enables the multi-year and inter-annual variability to be completely separated, and provides...' to read 'Decomposing the Yr1 hindcasts in this way provides a fair comparison because...' (now lines 269-273).

My second concern is related to the comparison with reanalysis. I wonder if it makes more sense to compare the model's results with reanalysis but using the same years / loads in the composites as in the model, this is, not using the GPCC precipitation time series to produce the wet and dry composites but the model's one. In this way you would be checking that the predicted mechanisms agree with the "observed" ones for those events that can be predicted by the system. You would remove from the possible causes of the mismatches the fact of using an independent dataset to determine the composites.

We understand the reviewer's concern - we had actually noted in lines 88-92 of the Supplementary Material that the use of GPCC data to compute composites may be the cause of some of the mismatches between the DePreSys3 and re-analysis composites.

To address this issue the reviewer provided the useful suggestion of using the DePreSys3 precipitation data to determine the wet and dry composites, which we have now done. Consequently we have replaced figures S2, S4 & S5 accordingly. Much of the results are very similar to those based on the GPCC time-series (now noted in lines 52-54 in Supplementary Material), reflecting the good rainfall skill of DePreSys3. Our discussion in section 3 has therefore not changed significantly, but has been modified accordingly to cater for the new results. In the main, this evaluation has improved the similarity between the model and re-analysis data, as predicted by the reviewer. For example, the circulation changes now very clearly show the difference between the northward migration versus the strengthening of the deep convection cell on multi-year and inter-annual timescales (Figures S2 c & d). The moisture flux analysis and moist static energy changes also now look better. On the other hand, the zonal circulation change on inter-annual timescales, although still showing an enhanced upwelling in the Sahel and a strengthened TEJ, does not look so similar to that of DePreSys3 (compare figures S2f and 3f). We suspect that this is a reflection that the main rainbelt in DePreSys3 is shifted to the south, as discussed in detail in the main article. We note this discrepancy in lines 45-47 in the Supplementary Material.

Details:

-line 10 “... multi-year (i.e. ~ 5 years) timescales ...”. I think it would be better to put “>” rather than “~”. The yr 2-5 average you use is like a running mean with a window length of 4 years, which is a low-pass frequency filter with an approximate cut-off period of 9 yrs.

Done

-line 62-64: “The fact that several models...robust over the historical period”. I'm not sure what this means: You were not sure about the statistical assessment you were using but once you applied and some models showed skill you are more confident in it ? I would remove this sentence.

We have removed this sentence

-line 222: “.. such as the Sahel Low..”. Do you mean Saharan Low?

Apologies – yes, we do mean the “Saharan Low’. We have corrected this instance and also in lines 143 and 152.

-Fig 3c: Make the colorbar the same as in the rest of wet minus dry composites for easier comparison.

Done

-Fig 5: where these calculated in the Sahel box? Please, clarify.

Yes. We have clarified this in the figure caption to read: ‘Anomalies between wet and dry Sahel summers of moist static energy (MSE) terms, averaged over the Sahel region.’

-Fig. 9: line 630: where is the upper level divergence calculated? Is ENSO also significantly anti-correlated with TEJ strength and upper level divergence in reanalysis? The sill map in plot c) is anomaly correlation? Please clarify.

To clarify, we have re-analysed the data to provide anomaly correlations for inter-annual timescales between the surface temperature within the ENSO region and the following: the zonal wind strength at 250 hPa (i.e, the TEJ), $r = 0.88$ (0.85, 0.90); the 250 hPa wind divergence in the Sahel box, $r = -0.57$ (-0.76, -0.44). We have also computed these for the re-analysis data and find them to be 0.65 (0.43, 0.77) between ‘ENSO’ and the TEJ and -0.58 (-0.80, -0.36) between ‘ENSO’ and upper level wind divergence. Correlations have been explained and noted in the figure caption.

The skill map in 9c & d is an anomaly correlation – we have clarified this in the figure caption.

Supplementary Information:

-Caption in table 1, last sentence: What do you mean by “Persistence skill is also included for DePreSys3? Typically persistence is calculated using observations, right? At least that was what I tried to suggest in the last review.

Apologies, that was a typo. The persistence skill was computed from the observations.

To clarify this, we have modified the table caption in the paper to read:

'Persistence skill is also included, computed by persisting the observed average over the equivalent number of summer seasons for the period prior to each model initialisation date.'

-Line 127: “In DePreSys3, we find that the AEJ does not ...” This would seem also the case for the reanalysis (Fig. S4), right?

Correct. We now state ‘or the re-analysis data’ in this line, and also refer the reader to Figure S4.

Reviewer #2 (Remarks to the Author):

The revisions made by the authors have substantially improved the paper. The clearer discussions, additional links to prior results, and testing other reanalysis and precipitation datasets improves the validity of the conclusions.

Thank you